# Psychological impacts from COVID-19 among university students: Risk factors across seven states in the United States

Matthew H. E. M. Browning[1]*, Lincoln R. Larson[2], Iryna Sharaievska[3], Alessandro Rigolon[4], Olivia McAnirlin[1], Lauren Mullenbach[2], Scott Cloutier[5], Tue M. Vu[6], Jennifer Thomsen[7], Nathan Reigner[8], Elizabeth Covelli Metcalf[7], Ashley D'Antonio[9], Marco Helbich[10], Gregory N. Bratman[11], Hector Olvera Alvarez[12]

1 Virtual Reality & Nature Lab, Department of Parks, Recreation and Tourism Management, Clemson University, Clemson, SC, United States of America, 2 Department of Parks, Recreation and Tourism Management, North Carolina State University, Raleigh, NC, United States of America, 3 Department of Parks, Recreation and Tourism Management, Clemson University, Clemson, SC, United States of America, 4 Department of City and Metropolitan Planning, The University of Utah, Salt Lake City, UT, United States of America, 5 Sustainability and Happiness Research Lab, School of Sustainability, Arizona State University, Tempe, AZ, United States of America, 6 Advanced Computing & Data Science, Clemson Computing & Information Technology, Clemson University, Clemson, SC, United States of America, 7 Department of Society and Conservation, W.A. Franke College of Forestry and Conservation, University of Montana, Missoula, MT, United States of America, 8 Recreation, Park, and Tourism Management Department, College of Health and Human Development, Pennsylvania State University, PA, United States of America, 9 Forest Ecosystems and Society, College of Forestry, Oregon State University, Corvallis, OR, United States of America, 10 Department of Human Geography and Spatial Planning, Faculty of Geosciences, Utrecht University, Utrecht, The Netherlands, 11 Environment & Well-Being Lab, School of Environmental and Forest Sciences, University of Washington, Seattle, WA, United States of America, 12 School of Nursing, Oregon Health & Science University, Portland, OR, United States of America

* mhb2@clemson.edu

**Data Availability Statement:** All relevant data are within the manuscript and its Supporting Information files.

## Abstract

### Background

University students are increasingly recognized as a vulnerable population, suffering from higher levels of anxiety, depression, substance abuse, and disordered eating compared to the general population. Therefore, when the nature of their educational experience radically changes—such as sheltering in place during the COVID-19 pandemic—the burden on the mental health of this vulnerable population is amplified. The objectives of this study are to 1) identify the array of psychological impacts COVID-19 has on students, 2) develop profiles to characterize students' anticipated levels of psychological impact during the pandemic, and 3) evaluate potential sociodemographic, lifestyle-related, and awareness of people infected with COVID-19 risk factors that could make students more likely to experience these impacts.

### Methods

Cross-sectional data were collected through web-based questionnaires from seven U.S. universities. Representative and convenience sampling was used to invite students to complete the questionnaires in mid-March to early-May 2020, when most coronavirus-related sheltering in place orders were in effect. We received 2,534 completed responses, of which 61% were from women, 79% from non-Hispanic Whites, and 20% from graduate students.

**Funding:** The author(s) received no specific funding for this work.

**Competing interests:** The authors have declared that no competing interests exist.

## Results

Exploratory factor analysis on close-ended responses resulted in two latent constructs, which we used to identify profiles of students with latent profile analysis, including high (45% of sample), moderate (40%), and low (14%) levels of psychological impact. Bivariate associations showed students who were women, were non-Hispanic Asian, in fair/poor health, of below-average relative family income, or who knew someone infected with COVID-19 experienced higher levels of psychological impact. Students who were non-Hispanic White, above-average social class, spent at least two hours outside, or less than eight hours on electronic screens were likely to experience lower levels of psychological impact. Multivariate modeling (mixed-effects logistic regression) showed that being a woman, having fair/poor general health status, being 18 to 24 years old, spending 8 or more hours on screens daily, and knowing someone infected predicted higher levels of psychological impact when risk factors were considered simultaneously.

## Conclusion

Inadequate efforts to recognize and address college students' mental health challenges, especially during a pandemic, could have long-term consequences on their health and education.

## 1 Introduction

A large number of studies support that the conclusion that the novel coronavirus (SARS-CoV-2) and its corresponding disease (COVID-19) have dramatically impacted people's mental health and behavior [1–5], with very few studies suggesting otherwise [6]. Mental health hotlines in the United States experienced 1,000% increases during the month of April, when most people were under lockdown because of the pandemic [7]. Some medical facilities have seen more deaths from suicide, presumably because of exceedingly poor mental health, than from COVID-19 infections [8]. Substance disorders in many people who were previously abstinent are expected to relapse during COVID-19, which will cause long-term economic and health impacts [9].

Although impacts are felt across populations—and especially in socially-disadvantaged communities and individuals employed as essential workers—college students are among the most strongly affected by COVID-19 because of uncertainty regarding academic success, future careers, and social life during college, amongst other concerns [10]. Even before the pandemic, students across the globe experienced increasing levels of anxiety, depressive moods, lack of self-esteem, psychosomatic problems, substance abuse, and suicidality [11]. Therefore, students may need additional resources and services to deal with the physical and mental health repercussions of the disease.

University administrators could best serve students if they better understood the impacts of COVID-19 and the risk factors of its psychological impacts. These impacts are of critical importance to warrant immediate mental health interventions focused on prevention and treatment [12]. Psychiatric and counseling services have historically been underutilized by college students [13, 14]. Understanding what subpopulations may suffer from unique combinations of psychological impacts may facilitate targeted interventions and successful treatment and coping strategies for individuals at greatest risk.

A recent review highlights some of the documented psychological impacts of COVID-19 on college students [15]. Many feel increased stress levels and anxiety and depressive symptoms as a result of changed delivery and uncertainty of university education, technological concerns of online courses, being far from home, social isolation, decreased family income, and future employment. These impacts have been observed in universities across the world [10].

Studies of the psychological impacts of COVID-19 on college studies in the United States, however, have been limited in their generalizability [10] due to examination of single institutions only [10, 16, 17]. We are aware of no studies that have been conducted with college students at multiple institutions across the United States during the pandemic. These schools collectively represent a somewhat unique context within higher education. The United States educates large numbers of students from around the world [18, 19]. Diverse student bodies may show different risk factors from more culturally-homogenous student bodies because of the diversity of value orientations [20] and sources of media consumption [16, 21–23]. Further, colleges in the United States cost more than higher education institutions nearly anywhere else in the world [24]; therefore, financial concerns may be particularly apparent in the United States. The United States also experienced the lowest global recovery rate from infection–in other words, the highest mortality rate post-infection–in the weeks leading up to the timing of the current study (April and May, 2020) [25]. This country continues to witness the highest incidence and mortality rates among Global North countries [26]. Such high rates aggravate the psychological impacts of the disease on infected and non-infected individuals [1].

In the current study, we investigate the psychological impacts of COVID-19 and associated risk factors on college students at seven universities across the United States. Our objectives are three-fold: 1) identify the array of psychological impacts COVID-19 has on students, 2) develop profiles to characterize students' anticipated levels of psychological impact during the pandemic, and 3) evaluate potential sociodemographic, lifestyle-related, and awareness of people infected with COVID-19 risk factors that could make students more likely to experience these impacts.

## 2 Methods

### 2.1 Study population

In spring 2020, 14,174 participants were recruited cross-sectionally from representative and targeted samples at seven large, state universities, which in sum enroll more than 238,000 students. Universities included Arizona State University in Tempe, AZ (approximately 52,000 undergraduate/graduate students enrolled in 2019); Clemson University in Clemson, SC (approx. 25,000); North Carolina State University in Raleigh, NC (approx. 34,000); Oregon State University in Corvallis, OR (approx. 29,000); Pennsylvania State University in State College, PA (approx. 54,000); University of Montana in Missoula, MT (approx. 11,000); and The University of Utah in Salt Lake City, UT (approx. 33,000). One institution (North Carolina State University) was able to utilize a university-wide representative sample. Other institutions used targeted samples in the home college(s) or department(s) of the corresponding author. Selection of sampling scheme (i.e., representative or targeted) was determined by human subject review board allowances and listserv availability. (Recruitment occurred over email listservs and course website announcements.)

This research was deemed exempt from the Clemson University Institutional Review Board. Also, all subjects provided written consent when they completed the online survey.

Recruitment started as soon as human subject approval was awarded and occurred over a two-to-three-week window at each institution. Because approval took longer at some

**Table 1. Sampling frames and recruitment windows across participating universities.**

| University | Recruitment Window | Sampling Frame | |
|---|---|---|---|
| | | *Description* | *N* |
| Arizona State University, Tempe, AZ | April 2 –April 10, 2020 | Undergraduate students in large-enrollment course in the School of Sustainability | 190 |
| Clemson University, Clemson, SC | March 17 –April 12, 2020 | All undergraduate and graduate students in the Department of Parks, Recreation and Tourism as well as all undergraduate students enrolled in large-enrollment introductory course in the Department of Communication | 1,168 |
| North Carolina State University, Raleigh, NC | March 26 –April 10, 2020 | Randomly sampled undergraduate and graduate students from across the university | 10,000 |
| Oregon State University, Corvallis, OR | April 30 –May 11, 2020 | All undergraduate and graduate students in the College of Forestry | 1,207 |
| Pennsylvania State University, State College, PA | April 27 –May 2, 2020 | All students in an undergraduate general education course offered by the Recreation, Park, and Tourism Management Department. | 141 |
| University of Montana, Missoula, MT | April 14 –April 27, 2020 | All undergraduate and graduate students in the College of Forestry and Conservation | 847 |
| The University of Utah, Salt Lake City, UT | March 26 –April 12, 2020 | All undergraduate and graduate students in the College of Architecture and Planning | 621 |
| | | **Total** | 14,174 |
| | | **Overall Response Rate** | 17.9% |

institutions, nationwide recruitment was staggered. No compensation for participation was provided. Sampling frames and recruitment windows are detailed in Table 1.

Of the 14,174 students invited to participate in the survey, we received 2,534 responses with data on most of the relevant variables; thus, this sample size was available for most of the descriptive statistics and bivariate associations. Missing/not reported data on race/ethnicity and gender occurred in approximately 11% of respondents. Therefore, complete data for multivariate analyses with all risk factors entered simultaneously—including race/ethnicity and gender—were available for 2,140 students. Table 2 provides the sample characteristics.

## 2.2 Measures

**2.2.1 Psychological impacts.** *2.2.1.1 Qualitative assessment.* We expected that it would be difficult to parsimoniously and comprehensively capture the broad array of impacts from COVID-19 on students with quantitative measures. Therefore, we utilized an open-ended questionnaire item that asked respondents, "We are interested in the ways that the coronavirus (COVID-19) pandemic has changed how you feel and behave. What are the first three ways that come to mind?" Three responses were required, and a fourth response was optional. This question was placed at the beginning of the questionnaire to avoid priming and order effects [27, 28].

*2.2.1.2 Quantitative assessment.* Regarding our selection of quantitative impacts to measure, we chose nine survey items based on information gathered from a review of previous research and new interview data. These nine survey items measured the following concepts: negative emotion states, preoccupation with COVID-19, feeling stressed, worry, and time demands.

Regarding the review of previous research, we examined studies of other large-scale disasters (i.e., the World Trade Center terrorist attacks on September 11, 2001; previous epidemics requiring quarantine), which are almost always associated with psychological impacts on the general population [29]. These studies provided some guidance on what impacts to measure for the impacts of COVID-19 on college students.

**Table 2. Characteristics of student respondents (N = 2,534)[a].**

| | | Range | |
|---|---|---|---|
| *Item* | *Mean (SD) / % (n)* | *Possible* | *Actual* |
| **Sociodemographic Factors** | | | |
| Female | 60.8% (n = 1502) | | |
| *Age* | | | |
| 18 to 24 | 76.8% (n = 1940) | | |
| 25 to 32 | 18.0% (n = 455) | | |
| 33 to 44 | 3.5% (n = 89) | | |
| 45 to 54 | 1.3% (n = 33) | | |
| 55 to 64 | 0.3% (n = 7) | | |
| 65 to 74 | 0.1% (n = 2) | | |
| *Race/Ethnicity* | | | |
| Non-Hispanic White | 78.6% (n = 1772) | | |
| Non-Hispanic Black | 4.2% (n = 95) | | |
| Non-Hispanic Asian | 12.8% (n = 289) | | |
| Hispanic | 4.3% (n = 98) | | |
| *Socioeconomic Status* | | | |
| Class (paternal) | 3.06 (1.03) | 1 (working class)– 5 (upper class) | 1–5 |
| Class (maternal) | 3.01 (1.00) | 1 (working class)– 5 (upper class) | 1–5 |
| Class (self) | 2.83 (1.02) | 1 (working class)– 5 (upper class) | 1–5 |
| Educational Achievement (paternal) | 4.55 (1.64) | 1 (less than high school) –7 (doctorate) | 1–5 |
| Educational Achievement (maternal) | 4.51 (1.50) | 1 (less than high school) –7 (doctorate) | 1–5 |
| Relative Income | 3.31 (1.11) | 1 (well below average)– 5 (well above average) | 1–5 |
| *Academic Status* | | | |
| Graduate Student | 19.8% (n = 499) | | |
| Undergraduate Student | 81.0% (n = 2053) | | |
| **Lifestyle-Related Factors** | | | |
| General Health | 3.31 (1.04) | 1 (Poor)– 5 (Excellent) | 1–5 |
| Body Mass Index (BMI) | 24.11 (4.67) | – | 5.35–60.08 |
| *Time Use (Last 24 Hours)* | | | |
| Screen time | 7.74 (2.81) | 0 hours– 12 hours | 0–12 |
| Outdoor time | 1.62 (1.70) | 0 hours– 12 hours | 0–12 |
| Exercise | 0.97 (1.15) | 0 hours– 12 hours | 0–12 |
| **COVID-19 Victim Awareness** | | | |
| Knowing Someone Infected | 24.6% (n = 591) | | |
| **Institutional Affiliation** | | | |
| Arizona State University | 6.1% (153) | | |
| Clemson University | 10.7% (271) | | |
| North Carolina State University | 58.3% (1473) | | |
| Oregon State University | 8.7% (219) | | |
| Pennsylvania State University | 4.7% (119) | | |
| University of Montana | 7.6% (193) | | |
| The University of Utah | 3.9% (98) | | |

[a]complete data for gender and race/ethnicity are available for 2,140 students.

Regarding new interview data, the corresponding author of the current study conducted unstructured interviews with adults on their experiences in the early stages of the COVID-19 pandemic. These interviews consisted of recruiting ten participants aged 18 years or older in

February 2020. Recruitment occurred in both low-risk and high-risk regions of the United States, including urban areas in Washington and rural areas in Tennessee, Iowa, and South Carolina. The interviews captured the feelings that interviewees experienced during the pandemic.

Negative emotion states comprised four of the survey items. Each item explained one of the basic negative emotions (i.e., being afraid, irritable, guilty, and sad) identified during the development of the positive and negative affect schedule (PANAS) [30]. Items were measured using the visual analogue scale (VAS) to provide data across a wide range of responses (1–100) with minimal participant burden [31]. Prompts asked respondents to indicate the extent to which they felt these things when they thought about the pandemic.

Preoccupation and feeling stressed comprised two more survey items. These were also measured with the VAS. Prompts once again asked respondents to indicate the extent to which they felt these things when they thought about the pandemic.

One more survey item measured worry—specifically anxious arousal. It was measured with a single item ("I worry a lot") from the Penn State Worry Questionnaire (PSWQ) that is strongly associated with the entire 16-item PSWQ, $r = 0.80$ [32]. Therefore, this single item succinctly captures the concept of worry/anxious arousal. A 5-point Likert-type agree-disagree response scale was used.

Two more survey items measured time demands. These were developed from survey prompts in the eating disorder literature [33]. Specifically, we asked to what extent respondents believed they spent a lot of time/thought on the pandemic, and to what extent they believed they spent *too much* time/thought on the pandemic. Once again, a 5-point Likert-type agree-disagree response scale was used.

The prompts for all nine of these survey items were delivered as reactions to the coronavirus rather than measures of general psychological states. Example include: "how stressed do you feel when you think about coronavirus," and "to what extent do you agree/disagree with the following: I worry about coronavirus all of the time."

**2.2.2 Risk factors.** Sociodemographic factors were self-reported and allowed identification of potential differences in impact levels by gender, age, race/ethnicity, socioeconomic status (SES), and academic status (undergraduate vs. graduate-seeking). SES was measured with perceived social class, which has been shown to accurately represent SES in student populations, using a battery of seven questions on class, parental education, and relative family income [34, 35]. To measure academic status, we asked respondents whether they were in pursuit of an undergraduate or graduate degree.

To account for possible lifestyle-related risk factors, we first considered general health factors such as general health status and body mass index (BMI). Health status was measured with a single item on respondents' "health in general" and a 5-point response scale (poor to excellent) [36]. BMI was calculated from self-reported height and weight. BMI has been implicated as a risk factor or confounder of the psychological impacts of COVID-19 [37, 38].

Another set of plausible lifestyle-related risk factors was time use. We utilized a recent recall question structure from the American Time Use Survey that strongly predicts objective time use and activity measures [39]. Three items were used to ask respondents to indicate how many hours they spent outdoors (at a park, on a greenway/trail, in a neighborhood/yard, etc.), in front of a screen (on a smartphone/computer, watching television, online gaming, etc.), and engaged in moderate or vigorous physical activity that caused an increase in breathing or heart rate (fast walking, running, etc.) in the past 24 hours [40, 41].

Regarding awareness of COVID-19 victims as a potential risk factor, we included two measures of knowing people who were diagnosed with the virus: someone in their family and someone in their community [42].

## 2.3 Analyses

To accomplish Objective 1, qualitative data from the open-ended responses were analyzed using content analysis with an inductive approach [43, 44]. Two independent researchers examined the data systematically to identify patterns and codes [43]. Each response was coded separately and reviewed for agreement [45, 46]. Interrater/intercoder agreement (kappa) score was 94.94% [47].

Objective 2 was accomplished in three steps. These included data imputations, data reduction, and profile identification.

We imputed missing values by bag imputation, which fits a machine learning regression tree model for each predictor as a function of all others [48]. In our dataset, 5.2% of the quantitative data were missing and imputed.

Next, we reduced the survey items related to levels of psychological impact into latent constructs using exploratory factor analysis (EFA) with oblimin rotation [49]. Scree plots and Very Simple Structure (VSS) criterion were used to identify the number of factors [50]. The VSS criterion evaluates the magnitude of the changes in goodness of fit with each increase in the number of extracted factors.

Last, using the composite scores from the EFA, we used the identified latent constructs from the psychological impact survey items as input variables in a latent profile analysis (LPA) [51]. Criteria for determining the number of profiles in the LPA included statistical adequacy of the solution and interpretability of each profile [52]. Indices used to determine statistical adequacy included the Akaike Information Criterion (AIC), Bayesian Information Criterion (BIC) and sample-size adjusted Bayesian Information Criterion [53]. For each of these indices, lower values represented better model fit. Also, the entropy criterion was calculated as a measure of classification precision [54]. We favored a parsimonious solution with fewer profiles over a more complex solution if this improved the interpretability of the LPA [53]. Z-scores of the input variables were used to interpret the profiles. The criteria to assign low and high values is not established and so we adopted previous studies' thresholds [53]. These included standardized scores between +0.5 and -0.5 being labelled as moderate, scores above 0.5 being labelled as high, and scores below -0.5 being labelled as low levels of psychological impact from COVID-19.

Objective 3 was achieved by modeling unadjusted (bivariate) and adjusted (multivariate) relationships between risk factors and profiles from the LPA. Unadjusted results are presented because multivariate models used a dichotomous outcome variable to distinguish students in the highest profile of psychological impact from those in the moderate or low profiles of impact (see Results for profile development and sample sizes within each profile). Determining risk factors for being in the high impact profile was deemed more important and actionable by university administrators than determining risk factors for each of the lower impact profiles, as would have been accomplished with a multinomial model. Thus, this modeling approach served a practical function; results could inform university administrators with tight budgets on how to prioritize funding for mental health interventions amongst students at greatest risk of high levels of psychological impact. Unadjusted results remained relevant, however, since they served the function of comparing risk factors between each level of impact profile in a simpler format than the output of a multinomial regression model.

For the unadjusted results, risk factors were evaluated with chi-squared contingency tables. Residuals from observed versus expected count comparisons determined the direction of the effect of the risk factors (i.e., more or less likely that a group was classified to a higher impact profile than another profile). Statistical significance of risk factors was calculated with Bonferroni adjustments to reduce Type I Error [55]. Continuous measures were reduced to

dichotomous or categorical factors based on clinically meaningful levels, past research, and data distributions. BMI was classified into four categories (less than 18 = underweight; 18 to 24.99 = normal; 25 to 29.99 = overweight; 30.0 and over = obese) [56]. General health was separated into two groups: poor/fair health and good/very good/excellent health [53]. Screen time was separated into less than eight hours on a device and eight or more hours on a device [57]. Time outdoors was split into three groups: Less than 1.00 hour, 1.00 to 1.99 hours, and 2.00 hours or more [58, 59]. Time spent exercising was also split into three groups: 30.00 minutes or less, 30.01 to 59.99 minutes, and 1.00 hour or more [60]. In addition, social class and relative income were split into three levels: below average, average, or above average. Levels of education were split into two levels: less than a 4-year college degree and a 4-year college degree or more [61].

For the adjusted results, we conducted generalized mixed-effects logistic regression to examine risk factors simultaneously and control for random (grouping) effects by institutional affiliation. To avoid collinearity in SES measures, whichever item correlated most strongly with psychological impacts was entered in the model. We used Variance Inflation Factor (VIF) values to test for multicollinearity. The proportion of variance explained was measured with conditional and marginal $R^2$ coefficients of determination [62–64]. Marginal $R^2$ represents the contribution of the predictors, which are modelled as fixed effects, whereas conditional $R^2$ accounts for the additional contribution of institutional affiliation (random effect) in addition to the fixed effects.

As a sensitivity analysis, we ran a logistic regression model with a subsample of respondents from the university that obtained a representative sample (North Carolina State University). This allowed us to evaluate the robustness of our nationwide sample, which otherwise utilized a convenience sampling approach.

Analyses were conducted in Excel for Mac Version 16.38 and R Version 3.6.2.

## 3 Results

### 3.1 Array of impacts

Qualitative data from the open-ended responses demonstrated a broad array of impacts from COVID-19 on college students' feelings (Table 3) and behaviors (Table 4). The most common changes in how students felt compared to before the pandemic were increased lack of motivation, anxiety, stress, and isolation. For example, one of the students reflected, "I'm normally extremely motivated, and I've never struggled with depression, but have recently felt very sluggish and melancholy." Another student described their feelings related to isolation as "I feel trapped. I don't have anywhere I need to go since I can't socialize, and I have schoolwork. But yet I still feel trapped due to actual restrictions and suggestions." The most frequent changes in student behavior compared to before the pandemic included more social distancing, more education changes, and less going out. Other concerning changes ranged from entrapment, boredom, fatigue, hopelessness, guilt, and inconvenience to hygiene, sleep, housing, employment, personal finances, and caretaking. For example, some students expressed their frustration with the financial situation, including one statement indicating: "I am BROKE. I lost my job because of this pandemic and now I can't pay for groceries." Other students were concerned about online learning. For example, one student commented: "I am constantly on edge about coursework: Did the computer register I submitted my exam? Did I see everything my teacher posted in Moodle? What happens if my internet goes out and I miss an assignment?"

Smaller numbers of students reported positive changes from the COVID-19 pandemic as well. These included optimism, productivity, adaptation, and empathy, as highlighted in the following quotes: "I've affirmed that people are capable of adapting in any circumstances" and "[I felt a] higher degree of empathy toward my community".

**Table 3. Open-ended responses on how COVID-19 changed how students feel.**

| Code | Description | % Sample Mentioned |
|---|---|---|
| Lack of motivation | Unmotivated/Hard to concentrate/Unproductive/Procrastinate more/Lazy | **21.5%** |
| Anxiety | Feeling anxious | **17.4%** |
| Stress | Overwhelmed | **14.6%** |
| Isolation | Lonely | **13.3%** |
| Worry | Worry about health of self or others, health, germs, other people's decisions, my own travel | 8.3% |
| Fear | Scared/paranoid/panicked | 8.1% |
| Entrapment | Limited/restricted/out of control/trapped/robbed | 6.7% |
| Boredom | Feeling bored | 6.2% |
| Uncertainty | Uneasy/directionless/confused/inquisitive/surreal | 6.0% |
| Sadness | Negative emotions | 5.8% |
| Depression | Expressing depressive feelings | 5.7% |
| Annoyance | Annoyed/Irritable/frustrated | 5.1% |
| Missing out | Disappointment | 5.1% |
| Mental health | Mental health is affected | 3.9% |
| Fatigue | Exhausted, lack of energy | 3.4% |
| Appreciation | Appreciate/grateful for life, friends, health, technology | 3.3% |
| Anger | Feeling angry | 1.8% |
| Relaxation | calm/relaxed | 1.6% |
| Optimism | Optimism/hopefulness/patient | 1.5% |
| Productivity | More productive/organized/creative | 1.4% |
| Hopelessness/ Helpless | Feeling hopeless/helpless | 1.3% |
| Adaptation | Flexible/adjustable to new situations | 1.0% |
| Empathy | Empathy towards others | 1.0% |
| Trust | Feeling trust/distrust of other people | 0.9% |
| Freedom | Independence, being in control | 0.8% |
| Lack of safety | Unsecure, precarious | 0.8% |
| Guilt | Guilt, shame, privilege | 0.5% |
| Overreaction / Underreaction | Primarily feeling overreaction/underreaction from others | 0.3% |
| Inconvenience | Feeling inconvenienced | 0.3% |

*Note*: Codes reported by 10% or more of the sample shown in **bold**.

## 3.2 Psychological impact profiles

Mean values of the psychological impact survey items are shown in Table 5. Eight of these were included in the EFA. (Feeling guilty demonstrated low communality ($h^2$ = .21) and was removed from further analyses.) All eight items displayed relatively normal distributions (S1 Fig). Criteria of the resulting model were acceptable: Tucker Lewis Index = 0.95; Kaiser-Meyer-Olkin (KMO) factor adequacy measure of sampling adequacy (MSA) = .89 [65]; significant Bartlett's test of sphericity, $\chi^2$(28) = 10503, $p$ < .001. The VSS Criterion [50] achieved a maximum of .93 with a two-factor solution, compared to .89 for a one-factor solution or .94 for a three-factor solution (S1 Table, S2 and S3 Figs). We labelled the first factor as "Emotional Distress" since it was composed largely of negative affect items (afraid, irritable, sad, preoccupied and stressed). The second factor was composed of three items dealing with how time was spent presumably in worry during the pandemic (worry, too much time and a lot of time), and so we labelled it "Worry Time." This is a term from clinical psychology that describes time spent reflecting on all the possible impacts of a health concern, including those worries that an

**Table 4. Open-ended responses on how COVID-19 changed how students behave.**

| Code | Description | % Sample Mentioned |
|---|---|---|
| Socializing (less/more) | Social distancing | **21.1%** |
| Education (change) | Change to online, no graduation, canceled school | **15.7%** |
| Going out (less) | More at home, less out, quarantining, less going to restaurants | **12.9%** |
| Exercise (change) | More/less exercise, being active | 9.5% |
| Eating pattern (change) | Hungry/Eating worse/Cook more/Not eating | 7.6% |
| Carefulness | Socially responsible, cautious | 7.5% |
| Hygiene (better) | Improved hand washing. Overall cleanliness | 7.4% |
| Sleep/ rest (change) | Sleep, lay on the couch | 6.9% |
| Schedule (change) | No schedule/Different schedule/Change in daily routines/change in planning/change in work-life balance | 5.9% |
| Housing (change) | Moved in with family members, moved to another location | 5.3% |
| Employment (loss) | Temporary or permanently lost job, income | 4.3% |
| Financial worry | Concerns about potential or real financial challenges | 4.2% |
| Employment (change) | Working from home, working less hours | 3.5% |
| Technology use (change) | Using less or more, avoidance | 3.4% |
| Shopping (change) | Less shopping, save money | 2.9% |
| Physical contact (less) | More aware of people | 2.7% |
| Time (more/less) | Perception of more or less time for activities | 2.6% |
| Travel (change) | Change in traveling plans/vacations | 2.0% |
| Outdoor activities (change) | Spending more or less time in the outdoors, outdoor activities | 2.0% |
| Self-reflection | Faith, self-reflection | 1.7% |
| News/ media | Reading more or less news | 1.6% |
| Leisure activity | Hobby/entertainment/projects | 1.4% |
| Physical health | Back pain, headache, sore throat, weight gain | 1.3% |
| Caretaking | Take care of children, older adults, while working/studying | 1.0% |
| Slow life | Slowing down | 0.5% |
| Driving (less) | Driving car less | 0.2% |

*Note*: Codes reported by 10% or more of the sample shown in **bold**.

individual cannot do anything about [66]. The internal reliability of the factors was high, Cronbach's α = .87 for Worry Time and .83 for Emotional Distress.

A three-profile solution fit the data best for the LPA. Information criteria decreased with additional profiles up to a five-profile solution, indicating a better model fit (S2 Table, S4 Fig). The elbow plot suggested minor improvements in model fit after a three-profile solution. Adding a fourth or fifth profile provided less interpretable results. Based on the combined information from the statistical criteria and interpretability, we retained a three-profile solution as our final model.

The three levels of psychological impact from COVID-19 resulting from the LPA are depicted in Fig 1. Positive z-scores indicate higher levels of impact and negative z-scores indicate lower levels of impact, compared to the average. Profile 1 ("high") represented students with higher than average levels of the two factors measuring psychological impacts (Emotional Distress, Worry Time) stemming from COVID-19. Profile 2 ("moderate") represented

**Table 5. Description of individual COVID-19 psychological impact survey items.**

| Impact | Survey Item | Mean (SD) | Range | |
|---|---|---|---|---|
| | | | Possible | Actual |
| Worry | "I worry about the coronavirus all of the time." | 4.01 (1.69) | 1 (Strongly disagree)– 7 (Strongly agree) | 1–7 |
| Too Much time | "I give too much time/thought to coronavirus." | 3.97 (1.75) | Same ranges as above | |
| Lot of Time | "I spend a lot of time thinking about coronavirus." | 4.34 (1.78) | | |
| Afraid | "How *afraid* do you feel when you think about coronavirus?" | 50.40 (27.78) | 0 (Not at all)– 100 (Extremely) | 0–100 |
| Irritable | "How *irritable* do you feel when you think about coronavirus?" | 59.44 (28.91) | Same ranges as above | |
| Sad | "How *sad* do you feel when you think about coronavirus?" | 60.98 (27.64) | | |
| Preoccupied | "How *preoccupied* do you feel when you think about coronavirus?" | 53.44 (27.40) | | |
| Guilty | "How *guilty* do you feel when you think about coronavirus?" | 24.15 (26.38) | | |
| Stressed | "How *stressed* do you feel when you think about coronavirus?" | 63.97 (26.96) | | |

students with moderate levels of the two factors, and profile 3 ("low") represented students with low levels of the two factors. Regarding profile membership, 45.2% of students (n = 1,146) were within the high impact profile, whereas 40.4% (n = 1,025) were in the moderate profile and 14.3% (n = 363) were in the low profile.

### 3.3 Risk factors

A summary of the risk factors with significant differences between impact profiles based on bivariate Chi-square tests is depicted in Fig 2. With respect to sociodemographic factors, women were more likely to be at risk than men ($\chi^2(2) = 88$, $p < .001$). Specifically, women were more likely to be in the high profile (residuals (RES) = 8.02, $p < .001$) and less likely to be in the moderate (RES = -2.75, $p = .036$) or low (RES = -7.54, $p < .001$) profile. Men demonstrated the opposite pattern. We did not observe differences by academic status ($\chi^2(2) = .3$, $p = .9$), although we did observe differences by age ($\chi^2(4) = 15$, $p = .005$). Students who were 18 to 24 years old were more likely to be in the moderate profile (RES = 3.81, $p = .0013$), and students who were 25 to 32 years old were less likely to be in the moderate profile (RES = -3.03,

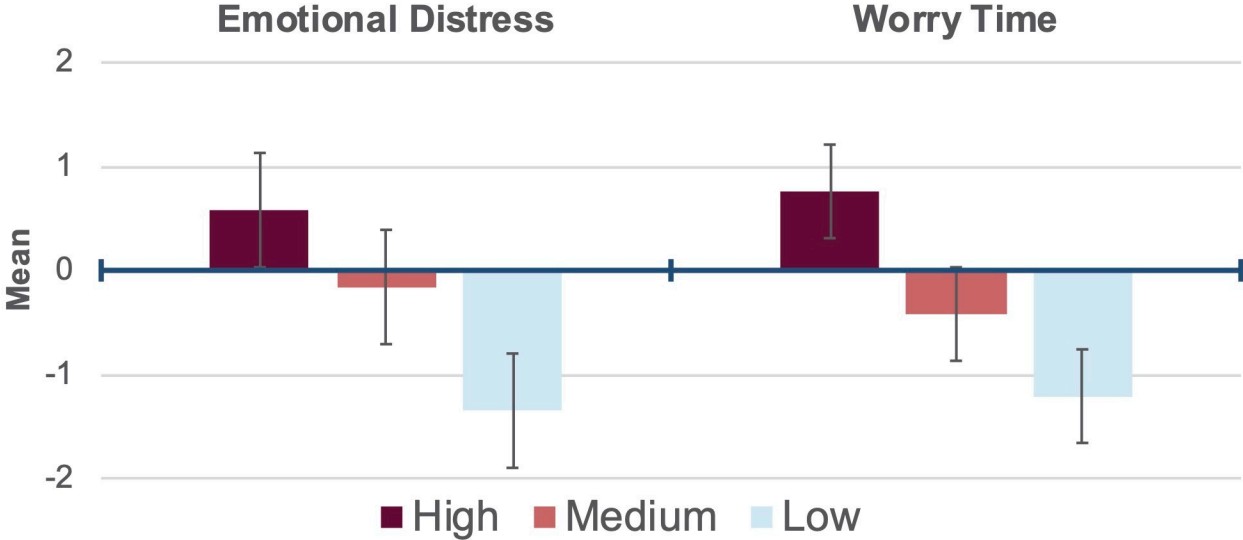

**Fig 1. COVID-19 psychological impact profiles derived from z-scores of eight items reduced to two factors using data from college students across the United States (n = 2,534).** Means and standard errors shown.

## A. Sociodemographic Risk Factors

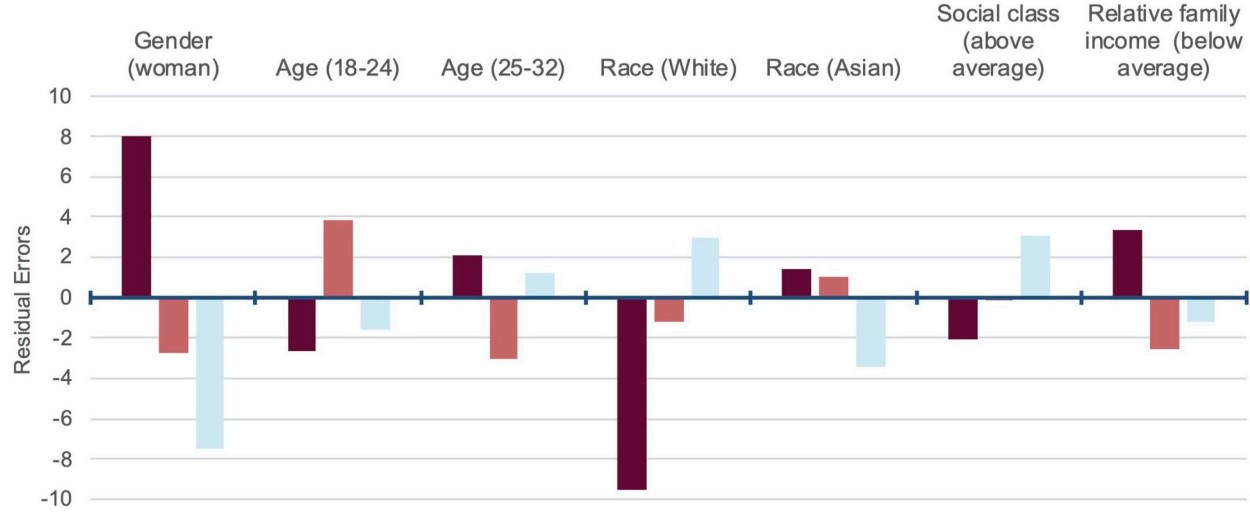

## B. Lifestyle-Related Risk Factors

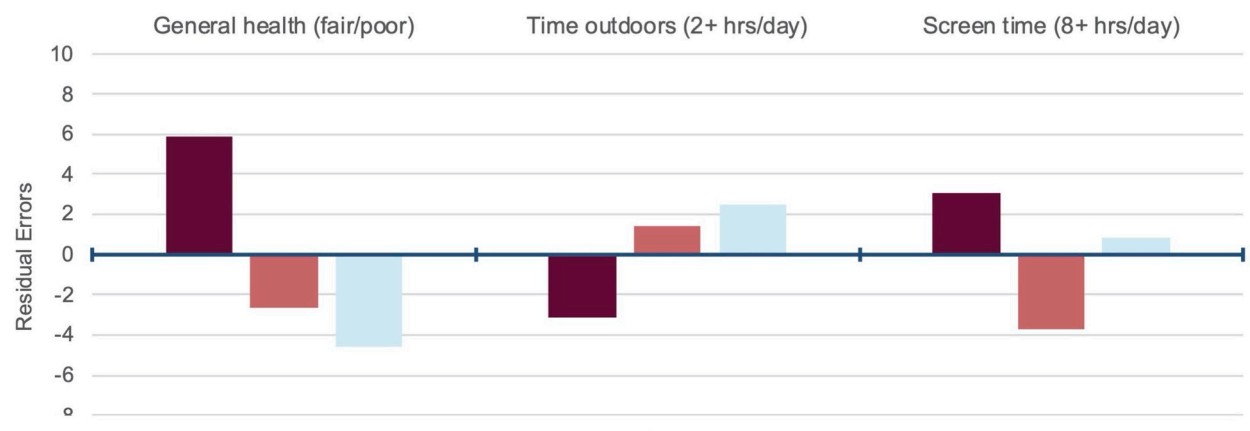

## C. COVID-19 Victim Awareness

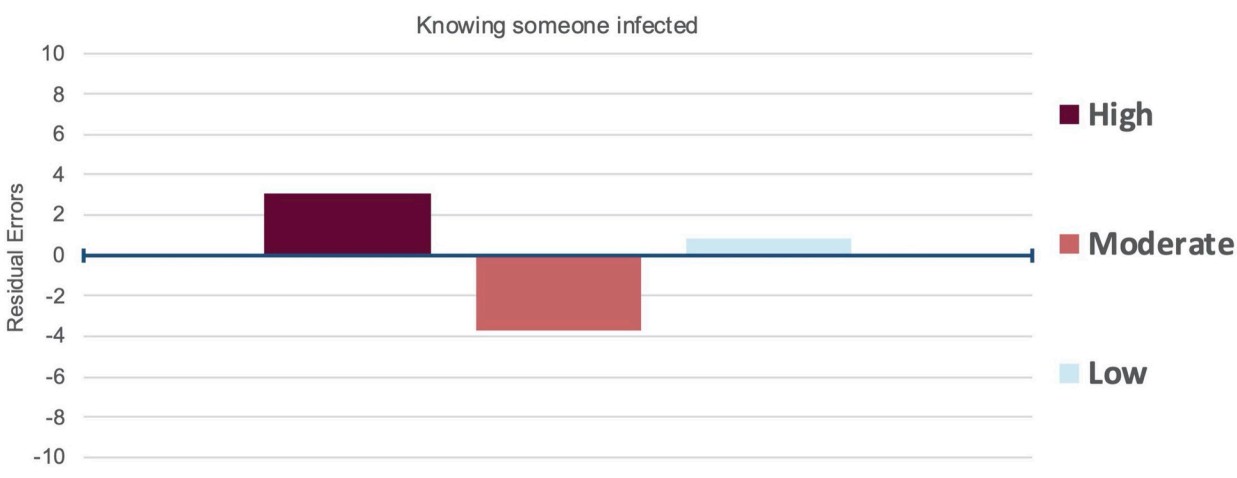

**Fig 2.** Sociodemographic (a), lifestyle (b), and COVID-19 victim awareness (c) risk factors associated with high, moderate, and low psychological impact profiles for students across the United States. Residuals from Pearson's chi-squared tests depict likelihood of profile membership based on risk factor. Only significant factors ($p < .05$) are reported. Reference groups include men; over 32 age; other race/ethnicity; average/above average SES (social class and relative family income); good/very good/excellent general health; less than 2 hours of time outdoors; less than 8 hours of screen time; and not knowing someone infected (COVID-19).

$p = .022$) than other profiles. No other significant differences between age groups by profile were found, $p > .05$.

We also observed racial/ethnic and SES differences in psychological impact levels. Specifically, we found differences by race/ethnicity ($\chi^2(6) = 18$, $p = .007$) with non-Hispanic Whites being more likely to be in the low profile (RES = 2.98, $p = .035$) and Non-Hispanic Asians being less likely to be in the low impact profile (RES = -3.42, $p = .0076$). No differences in impact profiles were observed for non-Hispanic Black students or Hispanic students, although sample sizes were small ($n = 95$ and 98, respectively). Parental educational achievement measures further showed no differences in profiles, $p = .5$ for maternal and .9 for paternal. No differences were observed for parental social class either, $p = .1$ for maternal and .2 for paternal. In contrast, student social class ($\chi^2(4) = 14$, $p = .008$), and relative family income ($\chi^2(4) = 14$, $p = .008$) differed by impact profile. Students who reported above-average social class were more likely to be in the low profile (RES = 3.07, $p = .019$), and students who reported below-average relative family income were more likely to be in the high profile (RES = 3.38, $p = .0065$). No other significant differences between ethnoracial groups or SES measures by profile were found, $p > .05$.

Lifestyle-related factors predicted differences in impact profiles. For instance, general health predicted assignment to different impact profiles ($\chi^2(2) = 41$, $p < .001$). Students with fair/poor health were more likely to be in the high profile (RES = 5.90, $p < .001$) and less likely to be in the moderate (RES = -2.67, $p = .045$) or low profile (RES = -4.58, $p < .001$). Students with good/very good/excellent health displayed the opposite pattern. No difference in impact profiles was observed for BMI ($\chi^2(6) = 9$, $p = .2$). We observed differences in impact profiles by time outdoors ($\chi^2(4) = 13$, $p = .01$) and screen time ($\chi^2(2) = 14$, $p = .001$) but not by exercise time ($\chi^2(4) = 6$, $p = .2$). Students who reported spending two or more hours outdoors were less likely to be in the high profile (RES = -3.17, $p = .014$), and students who reported spending more than eight hours on a device were more likely to be in the high profile (RES = 3.06, $p = .013$) and less likely to be in the moderate profile (RES = -3.67, $p = .0014$). Students spending less than eight hours on a device displayed the exact opposite trend. No other pair-wise comparisons in lifestyle-related factors were significant, $p > .05$.

Lastly, knowing someone who was infected with COVID-19 increased the likelihood of being at risk of psychological impacts ($\chi^2(2) = 14$, $p < .001$). Students who knew someone in their family or community who was infected were more likely to be in the high profile (RES = 3.06, $p = .013$) and less likely to be in the moderate profile (RES = -3.67, $p = .0014$). Students who did not know an infected person displayed the opposite pattern.

Five variables remained significant predictors of impact profiles in models adjusting for all risk factors simultaneously while controlling for institutional affiliation (Table 6). The SES measure entered in these models was social class of student, because it correlated more highly with psychological impact levels than other measures (S5 Fig). Students who were women, fair/poor general health, 18 to 24 years old, reporting 8 or more hours of screen time, and who knew someone infected with COVID-19 were more likely to be in the high profile. Non-Hispanic Asian students were marginally more likely to be in the high impact profile, $p = .091$. Effect sizes varied; women were approximately twice as likely to be assigned to the high impact profile as the moderate/low profile. Other predictors increased (or decreased) the likelihood of

**Table 6. Results of mixed-effects binary logistic regression modelling likelihood of risk factors predicting assignment to high COVID-19 psychological impact profile for students in seven United States universities ($N = 2,140$)** [a]**.**

|  | Log odds (95% CI) |
| --- | --- |
| Female | **2.018 (1.675, 2.431)** [***] |
| Age (18 to 24) | **1.375 (1.099, 1.719)** [**] |
| *Race/Ethnicity* |  |
| Non-Hispanic White | 0.868 (0.623, 1.208) |
| Non-Hispanic Asian | **1.274 (0.962, 1.687)** [^] |
| Class (Self) | 0.908 (0.802, 1.028) |
| General Reported Health | **0.619 (0.497, 0.772)** [***] |
| BMI | 1.036 (0.91, 1.178) |
| *Time Use (Last 24 Hours)* |  |
| Screen time | **1.216 (1.008, 1.466)** [*] |
| Outdoor time | 0.947 (0.834, 1.076) |
| Exercise | 1.039 (0.93, 1.161) |
| Academic Status (Graduate) | 0.946 (0.755, 1.186) |
| Knowing Someone Infected | **1.453 (1.173, 1.799)** [***] |
| Marginal $R^2$ / Conditional $R^2$ (%) | 6.8 / 7.3 |
| ICC | 0.005 |

Note

[^]$p < .10$

[*]$p < .05$

[**]$p < .01$

[***]$p < .001$. Predictors with $p < .10$ shown in bold. Due to identification with multiple races and missing values or other categories (i.e., mixed, prefer not to answer) for race/ethnicity and gender, 2,140 responses were available for multivariate analysis. These models were adjusted for random effects of institutional affiliation (S6 Fig), ICC = Intra-Class Coefficient.

being in the high impact profile by approximately 20% to 40%. No institutions emerged as significant random effects (S6 Fig). VIF values < 2.0 indicated no multicollinearity. Approximately 7% of the variance was explained by the predictors and institutional affiliations.

Sensitivity analyses with a subsample of respondents from the representative sample at North Carolina State University identified a similar set of predictors of psychological impact levels (S3 Table). Gender, age, general health, and knowing someone infected remained significant predictors. In contrast, screen time was no longer significant. Being Non-Hispanic Asian as marginally significant, $p = .070$, and social class was significant, $p = .0038$. Students of above average social class were 23.0% less likely to be assigned to the high impact profile.

## 4 Discussion

### 4.1 Key findings and interpretation of results

To evaluate the psychological impacts of COVID-19 on students in the United States, we collected over 2,500 survey responses from students at seven universities in late-February to mid-May 2020. Qualitative data from open-ended responses showed students experienced largely negative impacts of COVID-19 on psychological health and lifestyle behaviors. Among the most commonly reported changes were lack of motivation, anxiety, stress, and isolation, as well as social distancing, education changes, and going out less. Similar findings were reported by another study exploring the impact of COVID-19 on students at a single college in the

United States, revealing increases in sedentary lifestyle, anxiety, and depressive symptoms [16]. A global study examining experiences of students in 62 countries, including one university in the United States, found that students' expressed concerns about their academic and professional careers, as well as feelings of boredom, anxiety and frustration [10]. Increased anger, sadness, anxiety and fear were also reported by students in China [67]. Students in Switzerland reported a decrease in social interaction and higher levels of stress, anxiety, and loneliness [68]. More generally, adults have reported decreases in physical activity and food consumption increases during the COVID-19 pandemic quarantine compared to beforehand, as well as increases in binge drinking on average [69], which was identified in a small portion of our student respondents as well. Slight differences between our studies' results and results from studies conducted elsewhere may be due to the differences in student experience by geographical location. The United States is providing relatively little financial relief to college students during the pandemic compared to other Global North countries [70].

Quantitative survey measures captured the majority of the content that students entered in the open-ended responses (i.e., worry, stress, and fear) and informed the development of impact profiles. Students were assigned to one of three profiles—low (14% of the sample), moderate (40%) and high (45%)—based on the psychological impacts they reported experiencing in response to COVID-19.

In unadjusted models, students who were women, non-Hispanic Asian, in fair/poor health, of below-average relative family income, or someone who knew a family/community member infected with COVID-19 were at risk of higher levels of psychological impact. Students who were non-Hispanic White, above-average social class, spent at least two hours outside in the past day, or spent less than eight hours on screens in the past day were at less risk.

In multivariate models controlling, being a woman, being younger (18 to 24 years old), having poor/fair general health, reporting more screen time, and knowing someone infected were statistically significant risk factors. SES and identifying as non-Hispanic Asian were additional significant risk factors in the subsample of respondents obtained from representative sampling, whereas screen time was not significant in this sensitivity analysis.

These risk factor findings generally match those found in other studies that employed a case study approach within single United States universities/colleges. One longitudinal study of students at a public university in Nevada (n = 205) found that anxiety and depressive symptoms were greater in April 2020 than in prior months [17]. Women reported greater disruption to daily activities, mental and physical health, and personal finances than men. Contrary to our unadjusted findings, Asian or Asian-American students in the Nevada study reported lower levels of anxiety and depression than other races. A second longitudinal study with undergraduate students (n = 217) at a small liberal arts college in New Hampshire also found increases in anxiety, depression, and sedentary time during April 2020 relative to prior months [16]. COVID-19 risk factors for college students at other countries have been strikingly similar, as explained below.

Over ten studies, including several with college student populations, identify women as being at greater risk of psychological distress during the COVID-19 pandemic [1, 10, 21, 71–77]. Women are generally prone to depression and anxiety disorders [14], and although initial evidence indicated men were more susceptible to infection [77], our study supports the assertion that women appear to be more strongly impacted by the long-term psychological impacts of the pandemic. This observation may be attributable to higher levels of pre-existing psychopathology in women as well as gender differences in fear processing, which could translate to exacerbations of symptoms [78]. Also, male students tend to have higher confidence in the computer skills necessary for the transition to online course delivery [10]. Meanwhile, women are more concerned about impacts on their professional career and ability to study than men,

on average [10]. One study attributed these gender differences to greater emotional expression, less tolerance for uncertainty, and less-effective coping strategies amongst students who are women [75]. Women have also reported being more susceptible to "emotional hunger" and subsequent increased food intake than men during COVID-19 quarantine; these behaviors can lead to weight gain and poor mental health [73].

Our findings that fair/poor general health is a risk factor has been documented in numerous other populations during COVID-19 [79, 80]. In addition to comorbidity between mental and physical health status, people with pre-existing health problems and those with poor mental health show lower preparedness for disasters and suffer disproportionately more from disaster-related outcomes [81].

Several reasons explain our findings that younger students may be at greater risk than older students. Younger students (i.e., 18 to 24 years old, regardless of academic status) tend to be more worried about their future education and ability to pay for college education than older students [10]. Younger people also engage in social media more than older people during the pandemic [12, 82]. Given the dominance of the COVID-19 pandemic in the news, younger "always-on" students may be exposed to greater amounts of risk-elevating messages, which can lead to anxiety and poor mental health [16, 75].

Regarding our findings that non-Hispanic Asian students may be at greater risk than other races/ethnicities, several studies show higher psychological distress from COVID-19 in this population [10]. Asians and Asian Americans have reported being discriminated against by other students on social media during the pandemic [83]. Further, this population has experienced long-standing barriers to receiving and participating in mental health services [84].

The current study provides some support toward the mounting evidence that excessive screen time, including during the pandemic, may negatively impact mental health [85]. People who manage COVID-19 anxiety with excessive use of smartphones and other screen-based technology inadvertently learn more about the virus from the news, which fuels anxiety and ongoing coping through screens, thus causing a downward spiral [82]. Excessive use of digital media also detracts from time that could be spent on other health-promoting activities such as outdoor recreation [86]. Our study supports these relationships, suggesting negative impacts of screen time and positive impacts of "green time" on students' psychological health. The unadjusted analyses suggested that outdoor time predicted psychological impacts of COVID-19, although this variable was not significant in multivariate models. Other studies justify its consideration as a risk factor by university administrators. Both outdoor recreation [87] and nature exposure [88, 89] can improve psychosocial and eudaimonic well-being [90, 91]. Recent studies of people across the world show protective psychological effects of park and green space access during the pandemic [92] as well as lower rates of infection and mortality [93].

The finding that knowing someone infected is a risk factor for psychological impacts of COVID-19 is intuitive. Familiarity can increase the salience and perceived risk of becoming infected and dealing with subsequent health concerns, like COVID-19-related death [79]. Also, the threat of death from COVID has been associated with students' mental health and explainable by unhealthy levels of smartphone use [82].

As suggested in our unadjusted analyses and the multivariate model with the representative sample, SES may influence students' mental health during the pandemic. This might be a result of financial concerns affecting college students and their families [10]. SES has been documented as a predictor of COVID-19 fear and mental health concerns in other populations [10, 74, 79, 94–98]. Students coming from low-SES families may be more concerned about basic needs, like food and shelter, caused by loss of their or their parent's income [99]. Furthermore, since low-SES families are more susceptible to COVID-19 infection [98], students may be more concerned for their own and their families' safety.

## 4.2 Recommendations for universities

Given the large percentage of students assigned to the high psychological impact profile, universities would be well-served to address the mental health needs of their entire student body. Select programs that have promoted mental health—such as those at the University of Connecticut, University of Kentucky, and Northeastern—include virtual group exercise and meditation/mindfulness sessions, accountability buddies and exercise challenges and telemedicine/counseling visits [99]. These group meetings may be helpful not only in lowering anxiety but also in decreasing the sense of isolation reported by the students in this study. Digital interventions for students with clinical levels of anxiety or depression as well as potential for self-harm or suicide can involve automated and blended therapeutic interventions (such as apps and online programs), calls/text messages to reach those with less digital resources, suicide risk assessments, chatlines and forums, and other technologies to monitor risk either passively or actively [80]. Recently, Chen et al. [100] recommended a six-step intervention for the reduction in psychological impact risk amongst Chinese college students. These steps included the delivery of positive pandemic-related information, reduction in negative behavior, learning about stress management techniques, improvements in family relationships, increases in positive behavior, and adjustments in academic expectations.

Given the likelihood of ongoing psychological distress from COVID-19, universities may also consider helping students maintain healthy mindsets rather than avoiding stress [101]. In support of this proposition are recent findings that cognitive and behavioral avoidance (i.e., avoiding situations where exposure is possible and difficult thoughts about the pandemic) was the most consistent predictor of increased anxiety and depressive symptoms during the pandemic [17]. Cognitive reappraisal of stressful situations can alter their negative impacts [102, 103]. Training students to shift their educational experience mind-set to one that focuses on the "silver linings" and emerging opportunities may lead to "stress-related growth" and "toughening" [104, 105]. Adaptive mindsets can also help reorganize priorities to develop deeper relationships and greater appreciation of life [106], as well as help students to adjust to new ways of learning. Since a portion of the students in this study reported feeling less motivated, productive, and able to focus, switching to an adaptive mindset may help students persevere in their education and later in life. Finally, mindset reappraisals can increase well-being, decrease negative health symptoms, and boost physiological functioning under acute stress when a family member becomes infected or the pandemic creates rapid shifts in policies and procedures that affect students [107, 108].

Universities can further develop platforms that facilitate safe student social interaction. Many students seek out social interaction during their university experience [109–111]. However, as the findings of this study revealed, students' opportunities for socializing significantly decreased in the early stages of COVID-19. Missing "going out" and important milestone events (e.g., graduation, last sporting event) was a frequent response from our student participants. Other studies found that in order to maintain students' mental health during the first wave of the COVID-19 pandemic, they communicated online with close family members or roommates at least daily [10]. With college students, physical distancing does not and should not require "social distancing" [101]. Both synchronous (i.e., Zoom) and asynchronous (i.e., Facebook group) online interactions can foster bonding and bridging social connection [112–115], which can extend beyond social media posts and email listservs. Normal venues where people congregate such as places of worship, gyms, cafeterias, yoga studios and classes can be replicated online or even held outdoors in temperate weather on a schedule similar to what was in place prior to the pandemic [116]. Other recently-successful interventions include the facilitated online sharing of recipes, books, and podcasts as well as virtual movie, game, trivia,

or happy hour nights [99]. Providing support to student organizations to coordinate these virtual social activities could accelerate the availability of these resources.

Colleges and universities also have a moral obligation to boost their outreach to particularly vulnerable groups–that is, populations at risk of high levels of psychological impact from COVID-19 [14]. As documented in the impact profiles of our study, people at increased risk include women, younger students, students with pre-existing health concerns, students spending at least one-third of their day (including time spent sleeping) on screens, and students with family or community members who are infected with COVID-19. Monitoring and reporting rates of anxiety, depression, self-harm, suicide and other mental health issues within these groups is necessary to allocate counseling services and intervene pre-emptively and at times of acute symptomology [80]. Further, universities can provide accommodations for assignments and exams using a more personalized approach to learning and create enhanced opportunities for virtual social interactions with peers. These efforts may help at-risk groups succeed academically, build stronger relationships, and enhance their sense of belonging during distant learning [117].

Students in this study also expressed stress and anxiety associated with changes in education mode during the pandemics. As previous research has found, academic success may be supported with virtual town halls, regular email check-ins, virtual office hours, and peer mentoring [104]. Globally, students' satisfaction with university response to COVID-19 is predicted by students' satisfaction with pre-recorded videos during online course delivery, sufficient information on exams, satisfaction with teaching staff, satisfaction with websites and social media information with regular updates from the university, hopefulness, (lack of) boredom, (lack of) study issues, being on scholarship, being able to pay for school, and study discipline (social sciences tend to be less impacted than hard sciences or engineering) [10]. Universities may be encouraged by findings from another study on the switch to online courses; this study found many students were not challenged by the transition because of their aptitude toward digital learning and new technologies [118]. However, another study found new software platforms can be a challenge for some students [10].

## 4.3 Strengths and limitations

The primary strength of this study is the development of psychological impact profiles using data from universities across the United States. This sampling approach is also a limitation, however. Whereas all the included universities were teaching exclusively online during the study, their respective states and localities may have experienced differing levels of social distancing policy and enforcement. Another limitation related to the sample is the high percentage of non-Hispanic Whites. This occurrence was likely the result of the demographic composition of the colleges and departments targeted for recruitment [119]. Selection bias related to which students participated in the study questionnaire based on interest and access/availability is also possible [3].

Another limitation is the quantitative assessment of the psychological impacts of COVID-19, which could have limited the utility of our impact profiles. We did not measure substance abuse, which is expected to be a ramification of the virus [116] and which anxious individuals are prone to under-report [120]. Such counterproductive coping behaviors could be particularly problematic for college students [121]. Further, because our predictors explained a small amount of variance of the profiles, other unmeasured (or better measured) factors might predict students' psychological risk. For example, our single-item measures of leisure time activities could be improved with a more comprehensive assessment of time budgets such as those employed in episodic time use surveys [122].

We were primarily interested in reactions to the pandemic rather than how people were feeling/behaving during the pandemic. Therefore, we did not employ standardized measures of stress, anxiety, depression, or well-being. This limits our findings from being directly compared to other studies and pooled in meta-analyses.

Lastly, our measures were retrospective rather than longitudinal, which decreases our ability to say with confidence that the reported impacts were caused by COVID-19. However, we are fairly confident that the findings are attributable to the pandemic given our survey prompts. They specified students' responses to COVID-19 rather than asked generalized psychological states, and the findings strongly aligned with those of longitudinal studies of college students during the pandemic [16, 17, 37, 123–125].

## 5 Conclusion

Our cross-sectional study found that being a woman, being of younger age, experiencing poor/fair general health, spending extensive time on screens, and knowing someone infected with COVID-19 were risk factors for higher levels of psychological impact during the pandemic among college students in the United States. Unadjusted analyses also suggested that students who were non-Hispanic White, were not non-Hispanic Asian, were of higher-SES, or spent at least two hours outside experienced lower levels of psychological impact. That said, all students surveyed reported being negatively affected by the pandemic in some way, and 59% of respondents experienced high levels of psychological impact.

At the time that these data were collected, the education of over 1.5 billion students across the world were affected by COVID-19 [126]. Rates of student psychological distress were as high as 90% [17, 127]. Students must "Maslow before they can Bloom; " in other words, their basic physiological, psychological, and safety needs must be met prior to them focusing on–much less excelling–in academic life [99]. We recommend that university administrators take aggressive, proactive steps to support the mental health and educational success of their students at all times, but particularly during times of uncertainty and crisis–notably, the COVID-19 pandemic.

## Supporting information

**S1 Fig. Distributions and relationships between COVID-19 psychological impact survey items, including histograms, Pearson correlation coefficients, and scatter plots.**
(DOCX)

**S2 Fig. Diagram of EFA on COVID-19 psychological impact survey items.**
(DOCX)

**S3 Fig. Scree plot of EFA on COVID-19 psychological impact survey items.**
(DOCX)

**S4 Fig. Elbow plot of the information criteria for the latent profile analysis.**
(DOCX)

**S5 Fig. Correlations between socio-economic measures and the two psychological impact profiles.**
(DOCX)

**S6 Fig. Conditional mean values ("condval") and standard deviations of institutional affiliation (university) random effects from mixed-effects logistic regression predicting high versus medium/low psychological impact profile from COVID-19.**
(DOCX)

**S1 Table. Item loadings and fit statistics of EFA on COVID-19 psychological impact survey items.**
(DOCX)

**S2 Table. Fit indices, entropy and model comparisons for estimated latent profile analyses models.**
(DOCX)

**S3 Table. Results of binomial logistic regression modelling likelihood of risk factors predicting high versus low/moderate levels of COVID-19 psychological impact for students at North Carolina State University, where a representative sample was collected ($N$ = 1,312).**
(DOCX)

**S1 Data.**
(CSV)

## Author Contributions

**Conceptualization:** Matthew H. E. M. Browning, Lincoln R. Larson, Iryna Sharaievska, Alessandro Rigolon.

**Data curation:** Matthew H. E. M. Browning, Lincoln R. Larson, Iryna Sharaievska, Alessandro Rigolon, Olivia McAnirlin, Lauren Mullenbach, Scott Cloutier, Jennifer Thomsen, Nathan Reigner, Elizabeth Covelli Metcalf, Ashley D'Antonio.

**Formal analysis:** Matthew H. E. M. Browning, Iryna Sharaievska, Tue M. Vu.

**Investigation:** Matthew H. E. M. Browning, Lincoln R. Larson, Iryna Sharaievska.

**Methodology:** Matthew H. E. M. Browning, Lincoln R. Larson, Iryna Sharaievska, Alessandro Rigolon, Tue M. Vu, Nathan Reigner, Marco Helbich, Gregory N. Bratman, Hector Olvera Alvarez.

**Project administration:** Matthew H. E. M. Browning, Iryna Sharaievska.

**Resources:** Matthew H. E. M. Browning.

**Software:** Matthew H. E. M. Browning, Iryna Sharaievska, Marco Helbich.

**Supervision:** Matthew H. E. M. Browning.

**Validation:** Matthew H. E. M. Browning.

**Visualization:** Matthew H. E. M. Browning, Lincoln R. Larson, Tue M. Vu, Nathan Reigner.

**Writing – original draft:** Matthew H. E. M. Browning.

**Writing – review & editing:** Matthew H. E. M. Browning, Lincoln R. Larson, Iryna Sharaievska, Alessandro Rigolon, Olivia McAnirlin, Lauren Mullenbach, Scott Cloutier, Tue M. Vu, Jennifer Thomsen, Nathan Reigner, Elizabeth Covelli Metcalf, Ashley D'Antonio, Marco Helbich, Gregory N. Bratman, Hector Olvera Alvarez.

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
