## [Decision Letter · Decision Letter 0]

3 Sep 2020

PONE-D-20-24157

Psychological distress from COVID-19 among university students: Development of a risk classification scheme using data from seven states in the United States

PLOS ONE

Dear Dr. Browning,

Thank you for submitting your manuscript to PLOS ONE. After careful consideration, we feel that it has merit but does not fully meet PLOS ONE’s publication criteria as it currently stands. Therefore, we invite you to submit a revised version of the manuscript that addresses the points raised during the review process.

When reading the Reviewers’ comments, you will see that both Reviewers had concerns about how the analyses were conducted and/or reported. In my own review of the manuscript, I questioned the interpretation of both scree plots as indicating three-factor solutions. In both cases, only one eigenvalue exceeded the value of 1 and the drop-off in the plot could be interpreted to suggest a single-factor solution. Both Reviewers also recommended avoiding the term ‘Risk’ in naming groups identified by the latent profile analysis. Beyond the examination of risk factors or vulnerable groups to an adverse impact of COVID-19 (chi-square analyses), the authors might also consider including a multivariate analysis of these factors, rather than focusing exclusively on unadjusted analyses.

I would echo the concerns of Reviewer 1 regarding the short review of existing literature; there has been much research published on this topic in the last 6 months that would be informative to the reader. Both reviewers also had concerns about the recommendations made based on these data. I encourage you to consider the Reviewers' feedback and invite you to revise and resubmit your manuscript.

We look forward to receiving your revised manuscript.

Kind regards,

Christine M Wickens

Academic Editor

PLOS ONE

Journal Requirements:

Reviewers' comments:

Reviewer's Responses to Questions

**Comments to the Author**

1. Is the manuscript technically sound, and do the data support the conclusions?

Reviewer #1: Partly

Reviewer #2: Partly

2. Has the statistical analysis been performed appropriately and rigorously? 

Reviewer #1: I Don't Know

Reviewer #2: Yes

3. Have the authors made all data underlying the findings in their manuscript fully available?

Reviewer #1: Yes

Reviewer #2: No

4. Is the manuscript presented in an intelligible fashion and written in standard English?

Reviewer #1: Yes

Reviewer #2: Yes

5. Review Comments to the Author

Reviewer #1: The effects a pandemic has on the mental health of students are significant.

This study contains a lot of information, but the analysis performed and the interpretation of the data raises several questions. There are serious problems with the manuscript in its current form and I am not sure whether the study can be published on POne or not.

Regarding the introduction, a greater review of the scientific literature could be reflected. In fact, there are several current studies showing the effects of COVID lockdown on the mental health of college students in various countries.

A major problem is that the manuscript does not provide information on how the sample was specifically recruited (eg, type of sampling). Other important information from the sample is the mean age of the participants, the previous mental state and the taking of medication.

Regarding the interview and questionnaires, why not use validated instruments to assess mental health status? The use of different items generated ad hoc or selected from other forms does not guarantee the reliability of the measure extracted from them.

Regarding the exploratory factor analysis, nothing is indicated about the fulfillment of assumptions of normality, linearity or multicollinearity of the scores.

The main finding is that there are three different risk profiles for the students in the sample and several associated risk factors. However, the concept of risk factor does not seem appropriate. How to guarantee that other third variables are not those that account for the relationships found between these risk factors and the three profiles found?

Table 1 has an error. The total is not the sum of the participants presented in the column.

Finally, the recommendations made exceed the data found. I think they should adjust more to the results and limit themselves more to them

Reviewer #2: Summary and overall impression

This is an interesting and timely investigation of the impact of the first wave of the COVID-19 pandemic on university students in seven US states. The authors identified a range of psychological and behavioural COVID-19 related impacts which they used to classify participants into High, Moderate, and Low Impact groups (in the manuscript, the authors used the term ‘risk’ instead of ‘impact’). Latent profile analysis revealed that being a female, non-Latino Asian, and from a low SES were risk factors of inclusion in the High Impact group. Protective factors included being a White man.

I believe the main finding of this research is valuable in that it shows how the COVID-19 pandemic differentially impacts students along sociodemographic lines in a large and geographically dispersed US college student sample.

Specific areas for improvement - Major

I don’t believe the Behavioural Impacts listed in this study are appropriate given the pandemic context. Behaviour limiting may have more to do with state-issued directives (e.g., stay home orders) than personal choices to limit activities. For example, where I am, none of the behaviours listed are allowed because of state directives, not because of personal choice. During the period of data collection in each of the data locations, what were the COVID-19-related restrictions on these six activities? Had all study shifted online? Related to this, it appears participants had no opportunity to indicate whether the activities are something they would ever engage anyway (not everyone goes to the gym). Finally, in addition to weather, the time spent on activities in the past 24 hours would be highly variable based on local restrictions, the degree to which the participant’s study, work and other activities have shifted online, etc. For these reasons, I’d remove the Behavioural Impacts from consideration in this paper.

Single item factors are not interpretable. Two factor solutions should be accepted for the Psychological Impact items (and Behavioural Impact items, if you choose to keep them).

The authors have labelled the profiles High Risk, Moderate Risk, and Low Risk. These profiles would be more accurately labelled High Impact, Moderate Impact, and Low Impact. The sociodemographic, lifestyle, and COVID-19 characteristics are then Risk Factors for finding oneself in the High Impact group.

In the recommendations, the first two recommendations seem obvious but the third - targeted support for especially vulnerable groups – is more compelling and should be the focus of this section. The final recommendation seems impracticable given restrictions, social distancing, stay at home orders, etc.

Specific areas for improvement - Minor

The description of the analyses is spread across the Analyses and the Results section making it hard to follow. It should be presented as a single 'narrative' of the data analyses in the Results section.

I recognise that the authors’ expertise is in outdoor activities, but the inclusion of outdoor activity variables and literature feel ‘shoe-horned’ into the study and manuscript. I would remove references to outdoor activities and their contribution to psychological wellbeing as, in many cases, this is simply not an option.

There is an implication that screen time is a ‘bad thing’. During COVID-19, screens are how students access study, paid work, friends, and families. In this sense, increases in screen time is inevitable and desirable and should not be framed as the anti-outdoor activity in the way they might be in a non-COVID world.

The open-ended response tables probably belong in the supplementary materials.

The authors mention basic negative emotion states but don’t say what they are. What are they?

Why was BMI included?

Sub-headings would provide more structure to the manuscript (e.g., Psychological impacts, Behavioural impacts, Lifestyle risk factors, Sociodemographic risk factors, etc.).

Avoid using terms interchangeable - when talking about risk factors you also refer to them variables, characteristics, and attributes.

Participant numbers and characteristics should be included in the Participants section.

6. PLOS authors have the option to publish the peer review history of their article (what does this mean?). If published, this will include your full peer review and any attached files.

Reviewer #1: No

Reviewer #2: **Yes: **Francesca E. Collins

---

## [Author Response · Author response to Decision Letter 0]

25 Sep 2020

Reviewer 1

1.1. The effects a pandemic has on the mental health of students are significant. This study contains a lot of information, but the analysis performed and the interpretation of the data raises several questions. There are serious problems with the manuscript in its current form and I am not sure whether the study can be published on POne or not.

We appreciate your suggestions for improvement below and hope our revisions make the manuscript appropriate for publication in PLOS ONE.

1.2. Regarding the introduction, a greater review of the scientific literature could be reflected. In fact, there are several current studies showing the effects of COVID lockdown on the mental health of college students in various countries.

We conducted a forward search on the studies of college students previously listed in our paper, including the first article on the topic (Cao et al., 2020), as indicated by Grubic, Badovinac & Johri (2020). That first article has been cited more than 350 times. The forward search allowed us to expand our review of the extant literature and consider nearly 40 empirical and review articles on the impacts of COVID-19 amongst college students. Further, we considered the articles included in a systematic review on the psychological impacts of COVID on students, amongst other populations (Sankhi & Marasine, 2020). The updated review of the literature is briefly included in the introduction and extensively in the discussion.

Cao, W., Fang, Z., Hou, G., Han, M., Xu, X., Dong, J., & Zheng, J. (2020). The psychological impact of the COVID-19 epidemic on college students in China. Psychiatry research, 112934.

Sankhi, S., and Marasine, N. R. (2020). Impact of COVID-19 Pandemic on Mental Health of the General Population, Students, and Health Care Workers: A Review. preprints.org. doi:10.20944/preprints202007.0616.v1.

Grubic, N., Badovinac, S., and Johri, A. M. (2020). Student mental health in the midst of the COVID-19 pandemic: A call for further research and immediate solutions. Int J Soc Psychiatry 66, 517–518. doi:10.1177/0020764020925108.

1.3. A major problem is that the manuscript does not provide information on how the sample was specifically recruited (eg, type of sampling). Other important information from the sample is the mean age of the participants, the previous mental state and the taking of medication.

58% of the sample was from a representative sample of undergraduate students at one university (North Carolina State University, NCSU). Other university samples were convenience samples. 

To ensure the convenience samples did not bias our results, we reran all analyses with only the representative subsample (S1, Table A3). The findings remain essentially the same.

We agree that a lack of pre-COVID-19 data is a limitation of our study; however, we developed items to measure reactions to the coronavirus pandemic. These items were not intended for a repeated-measures design. We clarify this in the manuscript: 

[Methods] The prompts for all nine of these survey items were delivered as reactions to the coronavirus rather than measures of general psychological states. Example include: "how stressed do you feel when you think about coronavirus," and "to what extent do you agree/disagree with the following: I worry about coronavirus all of the time."

[Discussion] We were primarily interested in reactions to the pandemic rather than how people were feeling/behaving during the pandemic. Therefore, we did not employ standardized measures of stress, anxiety, depression, or well-being. This limits our findings from being directly compared to other studies and pooled in meta-analyses.

Regarding previous health status, our interest was on psychological impacts and symptoms, not on psychopathology as would have been indicated by medication use. We kindly remind the reviewer that we already include a general health status as well as BMI.

Table 1 now presents the age of participants within the categorical groupings that they were measured. Thank you for noticing that these were omitted. Age has also been considered as a risk factor in unadjusted and adjusted models.

1.4. Regarding the interview and questionnaires, why not use validated instruments to assess mental health status? The use of different items generated ad hoc or selected from other forms does not guarantee the reliability of the measure extracted from them.

We agree that previously validated items would have guaranteed reliability. However, as explained in our comment above, we were interested in reactions to the pandemic rather than how people were feeling/behaving during the pandemic. Our measures allowed us to capture responses to the pandemic using a cross-sectional study design. Standardized measures like GAD and PHQ would have allowed reporting the prevalence of depressive and anxiety symptoms. Without baseline measures, we would not have been able to determine how the pandemic changed the severity of these symptoms. Therefore, we believed using these survey items with prompts asking about reactions to the pandemic were appropriate for our study objectives.

That said, we now acknowledge that our measures limit our ability to directly compare results to other studies or pool our effects in future meta-analyses. We include this limitation in the manuscript:

[Discussion] We were primarily interested in reactions to the pandemic rather than how people were feeling/behaving during the pandemic. Therefore, we did not employ standardized measures of stress, anxiety, depression, or well-being. This limits our findings from being directly compared to other studies and pooled in meta-analyses.

1.5. Regarding the exploratory factor analysis, nothing is indicated about the fulfillment of assumptions of normality, linearity or multicollinearity of the scores.

We added histograms to show the relatively normal distributions of the survey items included in the EFA (S1, Figure A1). Also, we report on the lack of multicollinearity in the new multivariate regression model.

1.6. The main finding is that there are three different risk profiles for the students in the sample and several associated risk factors. However, the concept of risk factor does not seem appropriate. How to guarantee that other third variables are not those that account for the relationships found between these risk factors and the three profiles found?

To clarify our terminology, we now refer to the outputs of the latent profile analysis as three levels of psychological Impacts (thus, "Psychological Impact Profiles" or "Impact Profiles" for short). 

We believe the reviewer is concerned about unmeasured confounding. To address this concern, we report the results of a mixed-effects logistic regression with adjustments for all risk factors simultaneously entered in the model and random effects for institutional affiliation.

1.7. Table 1 has an error. The total is not the sum of the participants presented in the column.

Thank you for noticing this error. It has been corrected. We reviewed results in the other tables to double-check accuracy as well. No further changes were necessary.

1.8. Finally, the recommendations made exceed the data found. I think they should adjust more to the results and limit themselves more to them

We agree that the recommendation involving outdoor recreation exceeds the data found. It has been removed. 

Our recommendations are now more focused on specific findings from this study: changes in behavior and emotional state of students. Specifically, we provide recommendations on how university administration may lower students’ isolation by facilitating opportunities for socially-distanced interactions amongst students and their peers, between students and faculty, and between students and the larger campus community. We also provide recommendations on what techniques could be used to lower students’ level of anxiety and stress, as well as increase their motivation and general wellbeing. Lastly, we expanded our recommendations for working with vulnerable populations. 

Reviewer 2

2.1. This is an interesting and timely investigation of the impact of the first wave of the COVID-19 pandemic on university students in seven US states. The authors identified a range of psychological and behavioural COVID-19 related impacts which they used to classify participants into High, Moderate, and Low Impact groups (in the manuscript, the authors used the term ‘risk’ instead of ‘impact’). Latent profile analysis revealed that being a female, non-Latino Asian, and from a low SES were risk factors of inclusion in the High Impact group. Protective factors included being a White man. I believe the main finding of this research is valuable in that it shows how the COVID-19 pandemic differentially impacts students along sociodemographic lines in a large and geographically dispersed US college student sample.

Thank you for the nice summary.

2.2. I don’t believe the Behavioural Impacts listed in this study are appropriate given the pandemic context. Behaviour limiting may have more to do with state-issued directives (e.g., stay home orders) than personal choices to limit activities. For example, where I am, none of the behaviours listed are allowed because of state directives, not because of personal choice. During the period of data collection in each of the data locations, what were the COVID-19-related restrictions on these six activities? Had all study shifted online? Related to this, it appears participants had no opportunity to indicate whether the activities are something they would ever engage anyway (not everyone goes to the gym). Finally, in addition to weather, the time spent on activities in the past 24 hours would be highly variable based on local restrictions, the degree to which the participant’s study, work and other activities have shifted online, etc. For these reasons, I’d remove the Behavioural Impacts from consideration in this paper.

We agree behavioral limitations are suspect to bias and misinterpretation, since pre-COVID levels were not available and lockdown measures (and weather) dictate activities. We removed these variables from the manuscript entirely. 

We retain the activities in the last 24 hours because that question format mirrors the time diary approach of the American Time Use Survey. This asks about activity participation in the past 24 hours and correlates highly with objective measures (see https://www.bls.gov/tus/atususersguide.pdf and Tudor-Lock, Johnson & Katzmarzyk, 2010). The use of a time use measure with very recent reporting minimizes the recall bias common when questions reflect “average” behaviors over longer time horizons.

Tudor-Locke, C., Johnson, W. D., & Katzmarzyk, P. T. (2010). Frequently reported activities by intensity for US adults: the American Time Use Survey. American journal of preventive medicine, 39(4), e13-e20.

2.3. Single item factors are not interpretable. Two factor solutions should be accepted for the Psychological Impact items (and Behavioural Impact items, if you choose to keep them).

We now employ Very Simple Structure (VSS) criterion and scree plots to determine the number of factors. We find that a two-factor solution fits the psychological impact data best, and that both factors are represented by multiple survey items. Therefore, the single-item concern is no longer applicable.

Again, we removed behavioral impact items entirely.

2.4. The authors have labelled the profiles High Risk, Moderate Risk, and Low Risk. These profiles would be more accurately labelled High Impact, Moderate Impact, and Low Impact. The sociodemographic, lifestyle, and COVID-19 characteristics are then Risk Factors for finding oneself in the High Impact group.

We thank the reviewer for identifying this necessary change in terminology. We have made the necessary changes. We use "Psychological Impact Profiles" (not "Psychological Risks"). Again, behavioral items are removed based on your good feedback earlier. Sociodemographic, lifestyle-related, and awareness of infected individual characteristics are "Risk Factors".

2.5. In the recommendations, the first two recommendations seem obvious but the third - targeted support for especially vulnerable groups – is more compelling and should be the focus of this section. The final recommendation seems impracticable given restrictions, social distancing, stay at home orders, etc.

We agree that the recommendation involving outdoor recreation exceeds the data found. It has been removed. 

Our recommendations are now more focused on specific findings from this study: changes in behavior and emotional state of students. Specifically, we provide recommendations on how university administration may lower students’ isolation by facilitating opportunities for socially-distanced interactions amongst students and their peers, between students and faculty, and between students and the larger campus community. We also provide recommendations on what techniques could be used to lower students’ level of anxiety and stress, as well as increase their motivation and general wellbeing. Lastly, we expanded our recommendations for working with vulnerable populations. 

2.6. The description of the analyses is spread across the Analyses and the Results section making it hard to follow. It should be presented as a single 'narrative' of the data analyses in the Results section.

We reorganized the two sections to match another paper utilizing LPA and logistic regression recently published in PLOS ONE (Ekblom-Bak et al., 2020).

Ekblom-Bak, E., Stenling, A., Salier Eriksson, J., Hemmingsson, E., Kallings, L. V., Andersson, G., et al. (2020). Latent profile analysis patterns of exercise, sitting and fitness in adults – Associations with metabolic risk factors, perceived health, and perceived symptoms. PLOS ONE 15, e0232210–13. doi:10.1371/journal.pone.0232210.

2.7. I recognize that the authors’ expertise is in outdoor activities, but the inclusion of outdoor activity variables and literature feel ‘shoe-horned’ into the study and manuscript. I would remove references to outdoor activities and their contribution to psychological wellbeing as, in many cases, this is simply not an option.

This potential coping strategy was a focus given the increasing number of articles on the importance of greenspace/outdoor recreation during the pandemic, as well as debate over whether to keep public parks open (see citations below). However, since outdoor time no longer predicted psychological impact in multivariate models, we have minimized our discussion of it. 

Amerio, A., Brambilla, A., Morganti, A., Aguglia, A., Bianchi, D., Santi, F., et al. (2020). COVID-19 Lockdown: Housing Built Environment’s Effects on Mental Health. IJERPH 17, 5973–10. doi:10.3390/ijerph17165973.

Bell, M. L. (2020). Relationships between local vegetation level and human mobility patterns during COVID-19 for Maryland, USA. in, 1–1.

Dümpelmann, S. (2020). Urban Trees in Times of Crisis: Palliatives, Mitigators, and Resources. One Earth 2, 402–404. doi:10.1016/j.oneear.2020.04.017.

Dzhambov, A. M., Lercher, P., Browning, M., Stoyanov, D., Petrova, N., Nedkov, S., et al. (2020). Does indoor greenery provide an escape and support mental health during the COVID- 19 quarantine? Environ. Res., 1–39.

Gillis, K. (2020). Nature-based restorative environments are needed now more than ever. Cities & Health 00, 1–4. doi:10.1080/23748834.2020.1796401.

Gordon, L. (2020). Can virtual nature be a good substitute for the great outdoors? The science says yes. The Washington Post, 1–6. Available at: https://www.washingtonpost.com/video-games/2020/04/28/can-virtual-nature-be-good-substitute-great-outdoors-science-says-yes/.

Hanzl, M. (2020). Urban forms and green infrastructure – the implications for public health during the COVID-19 pandemic. Cities & Health 00, 1–5. doi:10.1080/23748834.2020.1791441.

Honey-Rosés, J., Anguelovski, I., Chireh, V. K., Daher, C., van den Bosch, C. C. K., Litt, J. S., et al. (2020). The impact of COVID-19 on public space: an early review of the emerging questions – design, perceptions and inequities. Cities & Health 00, 1–17. doi:10.1080/23748834.2020.1780074.

Kleinschroth, F., and Kowarik, I. (2020). COVID‐19 crisis demonstrates the urgent need for urban greenspaces. Frontiers Eco. Env. 18, 318–319. doi:10.1002/fee.2230.

Klompmaker, J. O., Hart, J. E., Holland, I., Sabath, M. B., Wu, X., Laden, F., et al. (2020). County-level exposures to greenness and associations with COVID-19 incidence and mortality in the United States. medRxiv, 1–14. doi:10.1101/2020.08.26.20181644.

Lu, Y., Zhao, J., Wu, X., and Lo, S. M. (2020). Escaping to nature in pandemic: a natural experiment of COVID-19 in Asian cities. osf.io, 1–35. doi:10.31235/osf.io/rq8sn.

McCunn, L. J. (2020). The importance of nature to city living during the COVID-19 pandemic: Considerations and goals from environmental psychology. Cities & Health 00, 1–4. doi:10.1080/23748834.2020.1795385.

Pouso, S., Borja, A., Fleming, L. E., Gómez-Baggethun, E., White, M. P., and Uyarra, M. C. (2020). Maintaining contact with blue-green spaces during the COVID-19 pandemic associated with positive mental health. osf.io. doi:https://doi.org/10.31235/osf.io/gpt3r.

Ramkissoon, H. (2020). COVID 19 Place Confinement, Pro-Social, Pro-environmental Behaviors, and Residents' Wellbeing: A New Conceptual Framework. fpsyg-11-02248.tex, 1–11. doi:10.3389/fpsyg.2020.02248.

Rice, W. L., Mateer, T. J., Reigner, N., Newman, P., Lawhon, B., and Taff, B. D. (2020). Changes in recreational behaviors of outdoor enthusiasts during the COVID-19 pandemic: analysis across urban and rural communities. Journal of Urban Ecology 6, 37–8. doi:10.1093/jue/juaa020.

Rousseau, S., and Deschacht, N. (2020). Public Awareness of Nature and the Environment During the COVID-19 Crisis. Environ Resource Econ 12, 1–11. doi:10.1007/s10640-020-00445-w.

Saadi, D., Schnell, I., Tirosh, E., Basagaña, X., and Agay-Shay, K. (2020). “There’s no place like home? The psychological, physiological, and cognitive effects of short visits to outdoor urban environments compared to staying in the indoor home environment, a field experiment on women from two ethnic groups.” Environ. Res., 109687. doi:10.1016/j.envres.2020.109687.

Samuelsson, K., Barthel, S., Colding, J., Macassa, G., and Matteo Giusti (2020). Urban nature as a source of resilience during social distancing amidst the coronavirus pandemic. OSF Preprint.

Slater, S. J., Christiana, R. W., and Gustat, J. (2020). Recommendations for Keeping Parks and Green Space Accessible for Mental and Physical Health During COVID-19 and Other Pandemics. Preventing Chronic Disease 17, 200204–8. doi:10.5888/pcd17.200204.

Stieger, S., Lewetz, D., and Swami, V. (2020). Psychological Well-Being Under Conditions of Lockdown: An Experience Sampling Study in Austria During the COVID-19 Pandemic. psyarxiv.com. doi:https://doi.org/10.31234/osf.io/qjhfp.

Understanding Psychosocial Factors Influencing Outdoor Recreation During the COVID-19 Pandemic (2020). Understanding Psychosocial Factors Influencing Outdoor Recreation During the COVID-19 Pandemic. Soc Nat Resour, 1–23.

Venter, Z., Barton, D. N., Gundersen, V., Figari, H., and Nowell, M. (2020). Urban nature in a time of crisis: recreational use of green space increases during the COVID-19 outbreak in Oslo, Norway. osf.io, 1–28. doi:https://doi.org/10.31235/osf.io/kbdum.

You, H., Wu, X., and Guo, X. (2020). Distribution of COVID-19 Morbidity Rate in Association with Social and Economic Factors in Wuhan, China: Implications for Urban Development. IJERPH 17, 3417–14. doi:10.3390/ijerph17103417.

2.8. There is an implication that screen time is a ‘bad thing’. During COVID-19, screens are how students access study, paid work, friends, and families. In this sense, increases in screen time is inevitable and desirable and should not be framed as the anti-outdoor activity in the way they might be in a non-COVID world.

We agree that increased screen time is inevitable. However, a growing body of literature suggests screen time is associated with access to unhealthy amounts of information and reactions to COVID-19 (see references below). Further, we find excessive screen time remains a significant predictor of psychological impacts in multivariate models. We maintain our reporting of it in the discussion.

Dubey, S., Biswas, P., Ghosh, R., Chatterjee, S., Dubey, M. J., Chatterjee, S., et al. (2020). Psychosocial impact of COVID-19. Diabetes & Metabolic Syndrome: Clinical Research & Reviews 14, 779–788. doi:10.1016/j.dsx.2020.05.035.

Duong, V., Pham, P., Yang, T., Wang, Y., and Luo, J. (2020). The Ivory Tower Lost: How College Students Respond Differently than the General Public to the COVID-19 Pandemic. arxiv.org

Hart, P. S., Chinn, S., and Soroka, S. (2020). Politicization and Polarization in COVID-19 News Coverage. Sci Commun, 107554702095073–19. doi:10.1177/1075547020950735.

Holmes, E. A., O'Connor, R. C., Perry, V. H., Tracey, I., Wessely, S., Arseneault, L., et al. (2020). Multidisciplinary research priorities for the COVID-19 pandemic: a call for action for mental health science. The Lancet Psychiatry 7, 547–560. doi:10.1016/S2215-0366(20)30168-1.

Huckins, J. F., daSilva, A. W., Wang, W., Hedlund, E., Rogers, C., Nepal, S. K., et al. (2020). Mental Health and Behavior of College Students During the Early Phases of the COVID-19 Pandemic: Longitudinal Smartphone and Ecological Momentary Assessment Study. J Med Internet Res 22, e20185. doi:10.2196/20185.

Liu, Q., Zheng, Z., Zheng, J., Chen, Q., Liu, G., Chen, S., et al. (2020). Health Communication Through News Media During the Early Stage of the COVID-19 Outbreak in China: Digital Topic Modeling Approach. J Med Internet Res 22, e19118–12. doi:10.2196/19118.

Stieger, S., Lewetz, D., and Swami, V. (2020). Psychological Well-Being Under Conditions of Lockdown: An Experience Sampling Study in Austria During the COVID-19 Pandemic. psyarxiv.com. doi:https://doi.org/10.31234/osf.io/qjhfp.

Xiong, J., Lipsitz, O., Nasri, F., Lui, L. M. W., Gill, H., Phan, L., et al. (2020). Impact of COVID-19 pandemic on mental health in the general population: A systematic review. J Affect Disord 277, 55–64. doi:10.1016/j.jad.2020.08.001.

2.9. The open-ended response tables probably belong in the supplementary materials.

Ordinarily, we might agree with the reviewer, but the available empirical evidence on the range of psychological and behavioral impacts of COVID-19 on students is relatively limited. Placing these tables in the supplementary materials would reduce their visibility and utility. Further, summarizing these findings in the main text without tables in the main text would increase in the overall length of the manuscript, which is already long. 

Ultimately, we are open to the PLOS ONE editor's suggestions on this, but if the journal allows, we trust that reporting these tables in a prominent location would ensure readers would appreciate easily finding them.

2.10. The authors mention basic negative emotion states but don’t say what they are. What are they?

We now identify these states in the manuscript:

Negative emotion states comprised four of the survey items. Each item explained one of the basic negative emotions (i.e., being afraid, irritable, guilty, and sad) identified during the development of the positive and negative affect schedule (PANAS).

2.11. Why was BMI included?

BMI has been implicated as a possible risk factor of psychological impacts from COVID-19 and a covariate in multivariate models predicting these impacts in past work (please see references below). We now justify the inclusion of BMI in the article using the below references.

Huang, Y., Wang, Y., Zeng, L., Yang, J., Song, X., Rao, W., et al. (2020). Prevalence and Correlation of Anxiety, Insomnia and Somatic Symptoms in a Chinese Population During the COVID-19 Epidemic. Front. Psychol., 1–9. doi:10.3389/fpsyt.2020.568329/full.

Xiao, J., Jiang, Y., Zhang, Y., Gu, X., Ma, W., and Zhuang, B. (2020). The Impact of Psychology Interventions on Changing Mental Health Status and Sleep Quality in University Students during the COVID-19 Pandemic. medRxiv. doi:10.1101/2020.09.01.20186411.

2.12. Sub-headings would provide more structure to the manuscript (e.g., Psychological impacts, Behavioural impacts, Lifestyle risk factors, Sociodemographic risk factors, etc.).

We appreciate this recommendation. We have added these sub-headings and hope they make the manuscript structure easier to follow.

2.13. Avoid using terms interchangeable - when talking about risk factors you also refer to them variables, characteristics, and attributes.

We thank the reviewer for identifying this inconsistency. We have made the necessary changes. 

2.14. Participant numbers and characteristics should be included in the Participants section.

We thank the reviewer for calling our attention to the fact that PLOS articles often have sample characteristics in the methods section. We have made this change.

---

## [Decision Letter · Decision Letter 1]

29 Dec 2020

Psychological impacts from COVID-19 among university students: Risk factors across seven states in the United States

PONE-D-20-24157R1

Dear Dr. Browning,

We’re pleased to inform you that your manuscript has been judged scientifically suitable for publication and will be formally accepted for publication once it meets all outstanding technical requirements.

Kind regards,

Chung-Ying Lin

Academic Editor

PLOS ONE

Additional Editor Comments (optional):

Reviewers' comments:

Reviewer's Responses to Questions

**Comments to the Author**

1. If the authors have adequately addressed your comments raised in a previous round of review and you feel that this manuscript is now acceptable for publication, you may indicate that here to bypass the “Comments to the Author” section, enter your conflict of interest statement in the “Confidential to Editor” section, and submit your "Accept" recommendation.

Reviewer #2: All comments have been addressed

2. Is the manuscript technically sound, and do the data support the conclusions?

Reviewer #2: Yes

3. Has the statistical analysis been performed appropriately and rigorously? 

Reviewer #2: Yes

4. Have the authors made all data underlying the findings in their manuscript fully available?

Reviewer #2: Yes

5. Is the manuscript presented in an intelligible fashion and written in standard English?

Reviewer #2: Yes

6. Review Comments to the Author

Reviewer #2: The authors have responded comprehensively to all reviewer concerns resulting in a far superior article.

7. PLOS authors have the option to publish the peer review history of their article (what does this mean?). If published, this will include your full peer review and any attached files.

Reviewer #2: **Yes: **Francesca Collins

---

## [Editor Report · Acceptance letter]

30 Dec 2020

PONE-D-20-24157R1 

Psychological impacts from COVID-19 among university students: Risk factors across seven states in the United States 

Dear Dr. Browning:

I'm pleased to inform you that your manuscript has been deemed suitable for publication in PLOS ONE. Congratulations! Your manuscript is now with our production department. 

Kind regards, 

on behalf of

Dr. Chung-Ying Lin 

Academic Editor

PLOS ONE